# Sequence Characteristics and Expression Analysis of *GhCIPK23* Gene in Upland Cotton (*Gossypium hirsutum* L.)

**DOI:** 10.3390/ijms231912040

**Published:** 2022-10-10

**Authors:** Maoni Chao, Jie Dong, Genhai Hu, Rui Ren, Ling Huang, Yanyan Li, Jinbao Zhang, Qinglian Wang

**Affiliations:** 1Postdoctoral Research Base, Henan Collaborative Innovation Center of Modern Biological Breeding, Henan Institute of Science and Technology, Xinxiang 453003, China; 2State Key Laboratory of Crop Biology, College of Agronomy, Shandong Agricultural University, Tai’an 271018, China; 3College of Agronomy, Henan Agricultural University, Zhengzhou 450046, China

**Keywords:** *CIPK23*, abiotic stress, expression pattern, function, *Gossypium hirsutum* L.

## Abstract

CIPK (calcineurin B-like-interacting protein kinase) is a kind of serine/threonine protein kinase widely existing in plants, and it plays an important role in plant growth and development and stress response. To better understand the biological functions of the *GhCIPK23* gene in upland cotton, the coding sequence (CDS) of the *GhCIPK23* gene was cloned in upland cotton, and its protein sequence, evolutionary relationship, subcellular localization, expression pattern and *cis*-acting elements in the promoter region were analyzed. Our results showed that the full-length CDS of *GhCIPK23* was 1368 bp, encoding a protein with 455 amino acids. The molecular weight and isoelectric point of this protein were 50.83 KDa and 8.94, respectively. The GhCIPK23 protein contained a conserved N-terminal protein kinase domain and C-terminal regulatory domain of the *CIPK* gene family member. Phylogenetic tree analysis demonstrated that GhCIPK23 had a close relationship with AtCIPK23, followed by OsCIPK23, and belonged to Group A with AtCIPK23 and OsCIPK23. The subcellular localization experiment indicated that GhCIPK23 was located in the plasma membrane. Tissue expression analysis showed that *GhCIPK23* had the highest expression in petals, followed by sepals, and the lowest in fibers. Stress expression analysis showed that the expression of the *GhCIPK23* gene was in response to drought, salt, low-temperature and exogenous abscisic acid (ABA) treatment, and had different expression patterns under different stress conditions. Further *cis*-acting elements analysis showed that the *GhCIPK23* promoter region had *cis*-acting elements in response to abiotic stress, phytohormones and light. These results established a foundation for understanding the function of *GhCIPK23* and breeding varieties with high-stress tolerance in cotton.

## 1. Introduction

Plants may encounter various stresses, such as extreme temperature, drought, nutrient deficiency, and pathogen infestation during their growth and development process. These stresses not only affect the yield and quality of plants, but also threaten the world’s food security. To cope with stresses, plants have established complex and effective regulatory mechanisms over their evolution [1,2,3]. For example, the intracellular level of calcium ions (Ca^2+^), which act as secondary messengers in plant cells, will appear as transient or persistent fluctuations in response to external stimuli [4]. Calcium signals are then sensed and transmitted by downstream calcium ion receptors, causing a series of intracellular biochemical and gene regulatory responses [5], so that plants can better adapt to the environmental stresses. 

CIPK proteins are a class of serine/threonine protein kinases and an important protein family in Ca^2+^-mediated plant signaling. They sense and transmit calcium signals through specific binding to calcineurin B-like proteins (CBLs) and play important roles in the growth and development and stress response of plants [6]. Since the Ca^2+^-mediated CBL–CIPK signaling system is a key regulatory node in the plant stress signal transduction pathway, the *CIPK* gene family has become a popular topic in plant stress physiology research in recent years. To date, the *CIPK* gene family has been identified and functionally analyzed in many species, including *Arabidopsis* [7], rice [8], maize [9], tomato [10], soybean [11], *Medicago* [12], and citrus [13]. In the *CIPK* gene family in *Arabidopsis*, the *AtCIPK23* gene has been studied deeply, and this gene is also considered to be the central hub for the *Arabidopsis* root system’s response to various stresses [14]. *AtCIPK23* regulates the drought tolerance of *Arabidopsis* plants by binding to *CBL1* and *CBL9* [15]. *OsCIPK23* is a homologous gene of *Arabidopsis AtCIPK23* in rice, and the inhibition of *OsCIPK23* expression by RNA interference can significantly reduce the seed setting rate of transgenic plants and result in a sensitive response of these plants to drought stress [16]. When *AtCIPK23* is mutated, *Arabidopsis* is sensitive to salt, and salt stress treatment significantly increases Na^+^, Ca^2+^, and reactive oxygen species (ROS) concentrations and root tip cell death [17]. In addition, *AtCIPK23* is also involved in the regulation of the uptake and transport of various mineral nutrients by plants. For example, AtCIPK23 regulates the activity of the downstream potassium transporter AKT1 by interacting with CBL1/CBL9 and promotes the uptake of potassium ions by *Arabidopsis* roots under low-K conditions [18]. This regulatory pathway has also been confirmed in rice [19] and cassava [20]. In addition, *AtCIPK23* has been found to have important physiological functions in the regulation of *Arabidopsis* in response to soil iron deficiency [21]. The *lks* mutant generated by the *AtCIPK23* mutation is highly sensitive to an iron-deficient environment, and the calcium-dependent CBL1/CBL9-CIPK23 pathway is involved in the response of *Arabidopsis* to iron deficiency [21]. The above results indicate that the *CIPK23* gene plays an important role in responding to various stress environments in plants, and the identification and cloning of the *CIPK23* gene in plants has great potential to help with the cultivation of crop varieties with high-stress tolerance.

Cotton is an economically important crop and is a potassium-sensitive crop, and its yield and quality are greatly affected by external environmental factors, such as low-temperature, salt, drought, etc. However, little work has been conducted on the functions of the *CIPK23* genes in upland cotton. In this study, based on the published genomic data of upland cotton [22], the *GhCIPK23* gene was cloned by homologous cloning, and its protein sequence characteristics, phylogenetic relationship, subcellular localization, tissue expression pattern, stress expression pattern, and *cis*-acting elements in the promoter region were analyzed. The results of this study can lay a theoretical foundation for further in-depth study of the function of *GhCIPK23* and the improvement of the stress tolerance of cotton through genetic engineering.

## 2. Results

### 2.1. Cloning of the Serine/Threonine Protein Kinase GhCIPK23 Gene in Upland Cotton

The *GhCIPK23* gene was highly homologous to the serine/threonine protein kinase *AtCIPK23* gene in *Arabidopsis*, which was identified by homologous cloning, and its gene ID was GH_D09G0913. Specific primers were designed based on the CDS of *GhCIPK23*, and using the cDNA of the roots of four-leaf stage cotton seedlings of Baimian No.1 as a template, the CDS of the *GhCIPK23* gene was amplified by PCR. The PCR product was detected by agarose gel, and a band with the same size as the CDS of *GhCIPK23* was obtained (Figure 1A). Further sequencing results showed that the PCR product was 1476 bp, of which the full-length CDS of *GhCIPK23* was 1368 bp (Figure 1B), and it was consistent with the reference genome sequence of upland cotton, indicating that the target gene was successfully cloned.

### 2.2. Protein Sequence Analysis of GhCIPK23 in Upland Cotton

Analysis of the GhCIPK23 protein sequence using the ExPASy online tools showed that this protein had 455 amino acids, and that the molecular weight and isoelectric point were 50.83 KDa and 8.94, respectively. In the amino acid composition of the GhCIPK23 protein, leucine (Leu) accounted for the highest proportion (10.3%), followed by lysine (Lys) at 9.0%, while tryptophan (Trp) accounted for the least proportion (0.9%). The analysis of the secondary structure of GhCIPK23 protein through the SOPMA online tools showed that alpha helix accounted for the highest proportion (38.02%), followed by random coils (32.75%) and extended strands (18.46%), while beta turn accounted for the least proportion (10.77%) (Figure 2).

### 2.3. Analysis of the Functional Domains of the GhCIPK23 Protein

Prediction analysis of the functional domains of GhCIPK23 using the SMART online tools showed that the protein had two conserved functional domains, namely, the protein kinase domain, located at 16 to 271 amino acids at the N-terminus, and the regulatory domain (with a NAF motif, which is the site of interaction between CIPK proteins and CBL proteins), located at 315 to 375 amino acids at the C-terminus (Figure 3A). Further multi-sequence alignment analysis showed that the GhCIPK23 protein had a high sequence identity with the CIPK23 proteins of other species, the highest identity (78.00%) with the *Arabidopsis* AtCIPK23 protein (Figure 3B) and that had the typical conserved features of the *CIPK* family, such as the ATP binding site, activation loop, NAF motif, and PPI motif (Figure 3C), which suggest that GhCIPK23 may have similar functions to CIPK23 proteins of other species.

### 2.4. Phylogenetic Analysis of GhCIPK23 Protein in Upland Cotton

To study the phylogenetic relationship between the GhCIPK23 protein in upland cotton and the CIPK proteins in other species, this study constructed a phylogenetic tree for a total of 60 CIPK proteins in three species, namely, *Arabidopsis thaliana*, *Oryza sativa* (rice), and *Gossypium hirsutum* L. (upland cotton). The phylogenetic tree showed that the plant CIPK proteins clustered into five subgroups, A-E, which contained 21, 4, 18, 13, and 4 CIPK proteins, respectively. The GhCIPK23 protein in upland cotton had the closest phylogenetic relationship to the AtCIPK23 protein in *Arabidopsis thaliana*, followed by the OsCIPK23 protein in rice, and belonging to Group A with AtCIPK23 and OsCIPK23 (Figure 4A). The nine *AtCIPKs* in *Arabidopsis* in subgroup A and the 11 *OsCIPKs* in rice have been confirmed to contain many introns in previous studies, and they belong to a class of intron-rich *CIPKs*. The *AtCIPKs* and the *OsCIPKs* in subgroups B to E do not contain or contain a few introns and belong to a class of intron-poor *CIPKs*. This study further analyzed the intron–exon structure of *GhCIPK23* and found that *GhCIPK23* in upland cotton, in subgroup A, similar to *AtCIPK23* and *OsCIPK23*, contained many introns. Specifically, the *GhCIPK23* and *AtCIPK23* both had 14 introns, while *OsCIPK23* had 13 introns (Figure 4B).

### 2.5. Subcellular Localization of GhCIPK23 Protein in Upland Cotton

*GhCIPK23* was cloned into the pROKII-GFP expression vector driven by the 35S promoter, with the empty vector expressing only green fluorescent protein (GFP) as a control, to determine the subcellular localization of GhCIPK23 protein using the tobacco transient transformation system. The results showed that green fluorescence was detected in the plasma membrane, cytoplasm, and nucleus of the empty vector, while green fluorescence was only detected in the plasma membrane of the expression vector containing *GhCIPK23* (Figure 5), indicating that GhCIPK23 is a protein that localizes to the plasma membrane.

### 2.6. Tissue Expression Analysis of GhCIPK23 Gene in Upland Cotton

The expression of *GhCIPK23* was found in all tissues of cotton tested by quantitative real-time PCR (qRT-PCR), but with different expression levels in different tissues (Figure 6). Specifically, *GhCIPK23* was expressed highest in petals, followed by sepals, roots, stems, and leaves and lowest in fibers (Figure 6), indicating that *GhCIPK23* may be involved in multiple processes in plant growth and development in upland cotton. In particular, it may play an important role in the formation and development of floral organs in upland cotton.

### 2.7. Analysis of the Response of GhCIPK23 Gene to Different Abiotic Stresses and ABA Treatment

Analysis of the expression pattern of *GhCIPK23* under drought, salt, low-temperature and exogenous ABA treatment showed that it was modulated in response to abiotic stresses and ABA treatment, but with different patterns under different stress conditions. *GhCIPK23* was significantly downregulated at 3 h of drought stress, began to increase later, and dropped significantly to its lowest level at 48 h of drought stress (Figure 7A). Under salt stress, the overall expression of *GhCIPK23* showed a significant decreasing trend (Figure 7B). Under low-temperature stress, the expression of *GhCIPK23* dropped to a minimum at 12 h, followed by an upregulation trend, reaching its highest level at 48 h (Figure 7C). In addition, *GhCIPK23* expression was also induced by exogenous ABA treatment, reaching its highest level at 3 h, and then started to decrease with time and decreased to the lowest level at 48 h (Figure 7D).

### 2.8. Cis-Acting Elements Analysis of the Promoter Region of GhCIPK23 in Upland Cotton

Further prediction analysis of the abiotic stress-responsive *cis*-acting elements in the 2000 bp upstream promoter region of *GhCIPK23* in upland cotton showed that the promoter region of this gene contained a low-temperature responsive *cis*-acting element (Figure 8). In addition, the promoter region of the *GhCIPK23* gene contained nine phytohormone-responsive type elements, including four MeJA-responsive elements (two TGACG motifs and two CGTCA motifs), two abscisic acid-responsive elements (ABREs), two salicylic acid-responsive elements (TCA elements), and one auxin-responsive element (TGA box), and 14 light-responsive types elements (Figure 8). These results indicated that the expression of *GhCIPK23* in upland cotton might be regulated by abiotic stress, phytohormone and light.

## 3. Discussion

Global climate change, increasing soil salinization, and water scarcity greatly challenge the world’s grain production. An in-depth study of the molecular regulatory mechanisms of plants in response to abiotic stress can improve the response of crops to changing environments and achieve high crop quality and yield, greatly contributing to national food security. Plant CIPKs are a specific class of serine/threonine protein kinases that can interact with the CBLs to form a complex signal network to sense and transmit calcium signals, playing an important role in plant response to stress [23,24]. *AtCIPK23* is a member of the *CIPK* gene family in *Arabidopsis* that is involved in the response to a variety of abiotic stresses and the uptake and transport of mineral nutrients [14]. However, knowledge about the *CIPK23* gene in upland cotton is much scarcer. In this study, the *GhCIPK23* gene in upland cotton was identified by homologous cloning. The full-length CDS of this gene was 1368 bp, encoding 455 amino acids, and the GhCIPK23 protein had a molecular weight of 50.83 KDa and an isoelectric point of 8.94. As important protein kinases, CIPKs have an N-terminal protein kinase domain and a C-terminal regulatory domain [25]. The NAF motif located in the C-terminal regulatory domain is necessary for the interaction between CIPK and CBL [26], and the PPI motif is the interaction site of CIPK with the protein phosphatase 2C (PP2C) [27]. These conserved domains are critical for the activation of CIPK proteins and their participation in stress responses. In this study, the GhCIPK23 protein in upland cotton contained a conserved N-terminal protein kinase domain and a C-terminal regulatory domain, which is consistent with the typical structural features of the *CIPK* gene family.

Further phylogenetic analysis showed that GhCIPK23 in upland cotton, AtCIPK23 in *Arabidopsis*, and OsCIPK23 in rice clustered into one branch and that the *CIPK* genes in this branch all contained many introns, while the *CIPK* genes in other branches contained few or no introns. Intron-poor branches are common among the *CIPK* genes [11,28,29,30], for example, the *CIPK* genes in the soybean cluster into intron-rich branches (subgroups I to III) and an intron-poor branch (subgroup IV) [11]. Similarly, the 22 *BvCIPKs* in sugar beet also phylogenetically cluster into five subgroups, of which the *BvCIPKs* in one subgroup have many introns (subgroup A) and the *BvCIPKs* in the other four subgroups (subgroups B to E) have one or no introns [30]. Subcellular localization analysis of GhCIPK23 protein showed that the protein was localized to the plasma membrane. In *Medicago*, most MsCIPK proteins are localized to the plasma membrane [12]. In sorghum, the CIPK proteins are localized to more positions, including the nucleus, mitochondria, plasma membrane, chloroplast, and cytoplasm [31]. CBL proteins can cause CIPK proteins located in different subcellular locations to relocate to the plasma membrane or vacuolar membrane to perform their functions. For example, the TaCIPK23 protein in wheat is localized to the cytoplasm, nucleus, and plasma membrane. When TaCBL1 is co-expressed with TaCIPK23 in wheat protoplasts, TaCBL1 can attach TaCIPK23 to the plasma membrane [32]. It is still unclear how GhCIPK23 and CBL proteins in upland cotton interact and where they localize to perform their functions; thus, further studies are needed.

Members of the *CIPK* gene family have different tissue expression patterns, indicating that they may have different functions [10,33,34,35]. For example, in tomato, *SlCIPK9* is mainly expressed in flowers; *SlCIPK1* is expressed in all tissues except roots; *SlCIPK5* is only expressed in roots, lateral roots, and seedlings; *SlCIPK2* and *SlCIPK19* are both expressed in all tissues tested; and *SlCIPK10* and *SlCIPK15* are expressed in tissues other than leaves [10]. In this study, we analyzed the tissue expression pattern of *GhCIPK23* in upland cotton and found that the expression of this gene was the highest in the petals. In *Arabidopsis*, the *AtCIPK23* gene is expressed in the roots, leaves, stems, flowers, and siliques of mature *Arabidopsis* plants, with the highest expression in the flowers [15]. The high expression of the *GhCIPK23* gene in the petals was consistent with the high expression of *AtCIPK23* in the flowers of *Arabidopsis*, suggesting that the two genes may have similar functions. A role for the *CIPK* gene in flower development has also been observed in rice and potato [16,33]. For example, *OsCIPK23* in rice is mainly expressed in pistils and anthers, and it is upregulated during rice pollination [16]. In potato, most *StCIPK* genes are highly expressed in petals and stamens, where they may play key roles in pollen germination [33]. In this study, the *GhCIPK23* gene was highly expressed in petals, which indicates that the gene may have important functions in the development of floral organs in upland cotton.

*CIPK* genes play important roles in the plant response to stress, and the CBL-CIPK signaling system composed of *CIPK* genes and upstream CBL proteins is involved in the response of plants to various stresses. In rice, the expression of 20 *OsCIPKs* is induced by abiotic stresses such as drought, salt, and low-temperature. Overexpression of *OsCIPK03*, *OsCIPK12*, and *OsCIPK15* could significantly improve the cold tolerance, drought tolerance, and salt tolerance of transgenic rice plants [36]. *AtCIPK23* responds to stresses such as drought, salt, and multiple nutrient deficiencies (including potassium deficiency, nitrogen deficiency, and iron deficiency) by relying on the CBL proteins, making it an important regulatory node for the cross-talk of various stress signals [14]. In this study, we found that the expression of *GhCIPK23* in upland cotton was affected to varying degrees under drought, salt stress, and low-temperature. Notably, *GhCIPK23* was significantly upregulated by low-temperature; thus, *GhCIPK23* may play an important role in the response of cotton to low-temperature stress. This result is consistent with the presence of one low-temperature responsive *cis*-acting element in the *GhCIPK23* promoter region. In addition, our study also found that drought stress induced the upregulation of *GhCIPK23*, which indicates that *GhCIPK23* may also play an important role in responding to drought stress. The important function of the *CIPK23* gene in response to drought stress has been reported in species such as *Arabidopsis*, rice, and wheat [15,16,32]. The expression of the *TaCIPK23* gene in wheat is induced by drought stress, and overexpressing *TaCIPK23* in wheat and *Arabidopsis* increases their germination rates, enhances their root systems, increases their osmotic pressure, and reduces their water loss rates under drought, thereby improving their survival rates [32]. However, our study found that *GhCIPK23* showed a different expression pattern under salt stress than under other abiotic stresses and that this gene was significantly downregulated within the salt stress treatment period. The same results were observed in poplar and wheat [37,38]. In poplar, the gene expression of *PtrCIPK1*, *PtrCIPK5*, *PtrCIPK8* and *PtrCIPK18* showed an overall downward trend under salt stress [37]. Under high salinity conditions, *TaCIPK25* expression was markedly downregulated in roots. Overexpression of *TaCIPK25* resulted in hypersensitivity to Na^+^, superfluous accumulation of Na^+^ and decreased salt tolerance in transgenic wheat [38]. Previous studies have shown that K^+^/Na^+^ homeostasis is the primary core response for plants to tolerate salinity [39]. The downregulated expression of *GhCIPK23* might participate in reducing Na^+^ accumulation and balancing K^+^/Na^+^ homeostasis under salt stress in upland cotton. Recent research shows that potassium signaling has been shown to act as a key determinant of plant salinity tolerance [40,41,42,43]. However, it is unclear whether *GhCIPK23* may also be involved in K^+^ uptake by plant roots under high sodium conditions, and the function of *GhCIPK23* in regulating Na^+^ and K^+^ homeostasis needs to be addressed in future research.

In hormonal responses, ABA plays a key role in plant response to various abiotic stress [44]. Previous studies have demonstrated that the expression of many *CIPK* genes is responsive to ABA treatment. For example, in potato, ABA induced upregulation of *StCIPK10* expression, and *StCIPK10* is a positive regulator of ABA-dependent responses [33]. In wheat, the expression of many *TaCIPKs* is responsive to ABA treatment [45]; overexpression of *TaCIPK23* can improve ABA sensitivity of plants and can regulate stomatal movement to reduce water loss, which also demonstrated that there is crosstalk between ABA signaling and drought [32]. In *Arabidopsis*, the expression of *AtCIPK3* is responsive to ABA, and it is shown to modulate ABA and low-temperature signal transduction [46]. Among the 33 *SiCIPK* genes detected in foxtail millet, the expression of 26 *SiCIPK* genes is induced by ABA, and some ABA-inducible *SiCIPK* genes are found to be responsive to multiple abiotic stresses [47]. Besides this, many studies also found that the ABA-responsive elements (ABREs) widely exist in the promoter region of *CIPK* family genes. In honeysuckle, ABREs are found in 13 out of 17 *LjCIPK* genes [34], which is also found in sugar beet (nearly one half of *BvCIPKs* is identified to contain ABREs) [30], suggesting that most *CIPK* genes might be involved in the ABA signal pathway. In our study, the expression of *GhCIPK23* was significantly upregulated by ABA treatment in upland cotton, which was consistent with the expression pattern of *OsCIPK23* [16] and *TaCIPK23* [32] under ABA treatment. In accordance with this, two abscisic acid-responsive elements (ABREs) were found in the promoter of the *GhCIPK23* gene in our study, which may explain why *GhCIPK23* expression was induced by ABA. These results suggest that *GhCIPK23* in upland cotton might be involved in the ABA signaling pathways.

The CBL-CIPK signaling system plays an important role in responding to low-potassium stress [19,48,49]. In *Arabidopsis* under low-potassium stress, the CBL1-CIPK23-HAK5 pathway can increase the potassium uptake in roots [49]. The CBL1-CIPK23 complex in rice regulates the potassium channel protein OsAKT1 to mediate the potassium uptake in rice roots under low-potassium stress [19]. In our previous studies, we found that the K^+^ transporter *GhHAK5* was specifically expressed in roots and is induced by low-potassium stress, but we have not clarified whether the CBL-CIPK23-HAK5 signaling pathway is widely present in upland cotton, much less whether *GhCIPK23* is involved in the regulation of *GhHAK5* expression to mediate K^+^ uptake under low-potassium stress, and we will investigate these regulatory mechanisms in a future study.

## 4. Materials and Methods

### 4.1. Experimental Materials

The experimental material was the upland cotton variety Baimian No.1, which was provided by the Institute of Cotton Research, Henan Institute of Science and Technology, and was planted in the experimental field. Under field conditions, the roots, stems, leaves, and flowers (petals and sepals) and 20 DPA (Day post anthesis, DPA) fibers of cotton plants were collected for tissue expression analysis.

### 4.2. Stress Treatment of Cotton Seedlings

Cotton seeds were sown in nutrient soil and cultured in the growth chamber until one true leaf stage and were transferred to 1/2 Hoagland’s nutrient solution for culture. After 7 days of growth, the seedlings with consistent growth were selected for treatment at 4 °C (low-temperature stress), 20% PEG 6000 (drought stress), 200 mM NaCl (salt stress) and 100 μM ABA (exogenous abscisic acid treatment). The roots of cotton seedlings were sampled at 0 (CK), 3, 6, 12, 24, and 48 h of treatment for stress expression analysis.

### 4.3. RNA Extraction and cDNA Synthesis

The total RNA from different cotton tissues was extracted according to the manual of the plant RNA extraction kit (Code No.: DP441, TIANGEN, Beijing, China). The reverse transcription kit (Code No.: 6210A, TaKaRa, Dalian, China) was used for reverse transcription, and a total of 2 μg RNA was used in reverse transcription experiments, and 20 μL cDNA was obtained. The synthesized cDNA was stored at −20 °C for future use.

### 4.4. Amplification of the CDS Sequence of the GhCIPK23 Gene

The specific primers for cloning the *GhCIPK23* gene are as follows: *GhCIPK*23-F (5’-3’): GGATAGACAACGATGGCGACT; *GhCIPK23*-R (5’-3’): TTCAACAAACCAACATAAAAA. Using the cDNA of the roots of four-leaf stage cotton seedlings as the template, the PrimeSTAR GXL DNA Polymerase (Code No.: R050A, Takara, Dalian, China) was used for PCR amplification, and the PCR reaction system was carried out according to the instructions. The PCR program was 94.0 °C (pre-denatured) for 2 min; 98.0 °C (denaturation) for 20 s and 56.0 °C (annealing and extension) for 1 min 30 s, for 33 cycles; and 72 °C (final extension) for 10 min. After the PCR product was purified, a poly A-tail was added, and the product was ligated into the pMD19-T cloning vector to transform E. coli *DH5α* competent cells. The single clones were randomly selected for PCR identification and sent to Shenzhen BGI for sequencing.

### 4.5. Protein Sequences Analysis

Using ExPASY (https://web.expasy.org/protparam/, accessed on 8 April 2022) [50] to predict and analyze the amino acid length, molecular weight, isoelectric point, and amino acid composition of GhCIPK23 protein. Functional domains of GhCIPK23 protein were analyzed using SMART (http://smart.embl-heidelberg.de/, accessed on 9 April 2022) [51] online tools. The SOPMA (https://npsa-prabi.ibcp.fr/cgi-bin/npsa_automat.pl?page=npsa_sopma.html, accessed on 9 April 2022) [52] was used to predict the secondary structures of GhCIPK23 protein. The protein sequence identity analysis was completed by BioEdit 7.1.11 software [53].

### 4.6. Construction of Phylogenetic Tree

The protein sequences of the *CIPK* gene family of *Arabidopsis* and rice were downloaded from the UniProt protein database (https://www.uniprot.org/, accessed on 2 May 2022). The Neighbor-Joining (NJ) method in MEGA 5.02 software [54] was used to construct a phylogenetic tree. The parameter settings were as follows: bootstrap value was set to 1000, the p-distance model was selected, and pairwise deletion was used for processing missing data.

### 4.7. Gene Structure Analysis

The genomic sequences and CDS sequences of *Arabidopsis AtCIPK23* (At1g30270) and rice *OsCIPK23* (LOC_Os07g05620) were obtained from the Phytozome website (https://phytozome-next.jgi.doe.gov/, accessed on 6 May 2022). The schematic diagram of the intron-exon structure of the gene was drawn using Gene Structure Display Server (GSDS) (http://gsds.gao-lab.org/, accessed on 6 May 2022) [55] online tools.

### 4.8. Subcellular Localization

The pROKII-GFP vector was double digested using the restriction enzymes *BamH* I and *Kpn* I. The specific primers were designed to amplify the *GhCIPK23* gene, and the specific primers were as follows: GhCIPK23-GFP-F (5’-3’): gggactctagaggatccATGGCGACTCGCAGTAGCGGTT; GhCIPK23-GFP-R (5’-3’): ttgctcaccatggtaccTGAGGAGCCTGCCGCAGCACT. The pROKII-GhCIPK23-GFP vector was constructed by homologous recombination. The constructs were transformed into GV3101 *Agrobacterium* competent cells. The signals were observed using a laser confocal microscope (Zeiss LSM780).

### 4.9. Quantitative Real-Time PCR (qRT-PCR) Analysis

qRT-PCR amplification was performed by the SYBR Green dye method with *Actin* as a reference gene. Specific primers were as follows: *GhCIPK23*-q-F (5’-3’): TACACCCAAATGTCATCCGC; *GhCIPK23*-q-R (5’-3’): TGTCAAAAAGTTCACCACCAGT; *Actin*-q-F (5’-3’): GACCGCATGAGCAAGGAGAT; *Actin*-q-R (5’-3’): GCTGGAAGGTGCTGAGTGAT. The qRT–PCR system contained 10 μL of 2×SYBR Premix Ex Taq (Code No.: RR820 L, Takara, Dalian, China), 0.8 μL of 10 mmol L^−1^ forward primer, 0.8 μL of 10 mmol L^−1^ reverse primer, 1 μL of cDNA, and 7.4 μL of ddH_2_O. The amplification program was 95 °C for 30 s for 1 cycle and 95 °C for 5 s and 60 °C for 20 s for 40 cycles. Three replicates were set for each sample, and the relative expression levels of genes were calculated using the 2^−ΔΔCt^ method.

### 4.10. Cis-Acting Elements Analysis in the Promoter Region

The 2000 bp promoter sequence upstream of *GhCIPK23* was obtained from the CottonFGD website (https://cottonfgd.org/, accessed on 16 May 2022) [56]. Then, the *cis*-acting elements in the promoter region were analyzed using the PlantCare (http://bioinformatics.psb.ugent.be/webtools/plantcare/html/, accessed on 16 May 2022) [57] online tools.

## Figures and Tables

**Figure 1 ijms-23-12040-f001:**
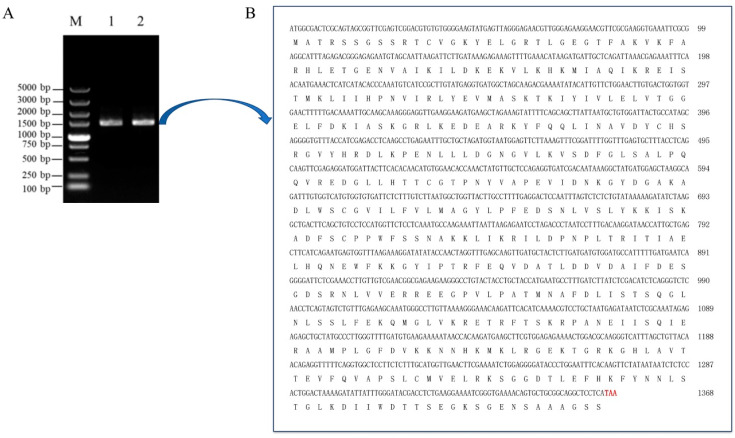
Cloning of the CDS sequence of *GhCIPK23* gene in upland cotton. (**A**) Agarose gel electrophoretic analysis of PCR amplification product of *GhCIPK23* gene; 1 and 2: Target gene; M: DNA marker (DL5000). (**B**) The CDS sequence of *GhCIPK23* and its encoded protein by sequencing. the red base represents the stop codon.

**Figure 2 ijms-23-12040-f002:**
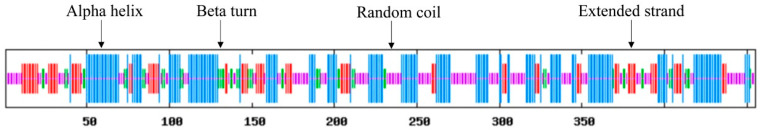
Secondary structure analysis of GhCIPK23 protein in upland cotton. The blue, green, purple and red lines represent alpha helix, beta turn, random coil and extended strand respectively.

**Figure 3 ijms-23-12040-f003:**
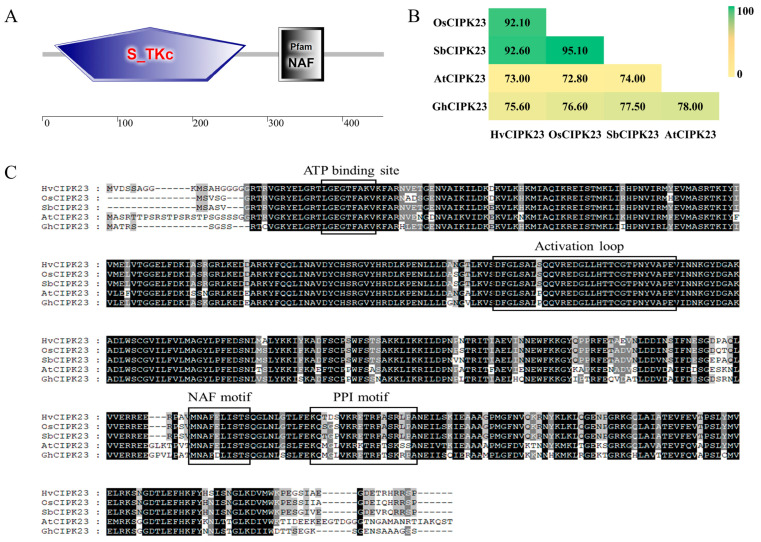
Domain and sequence identity analysis of GhCIPK23 protein in upland cotton. (**A**) Protein domain prediction of GhCIPK23. (**B**) Protein sequence identity analysis of GhCIPK23 and CIPK23 protein of other species. (**C**) Amino acid sequence alignment of GhCIPK23 and CIPK23 protein of other species.

**Figure 4 ijms-23-12040-f004:**
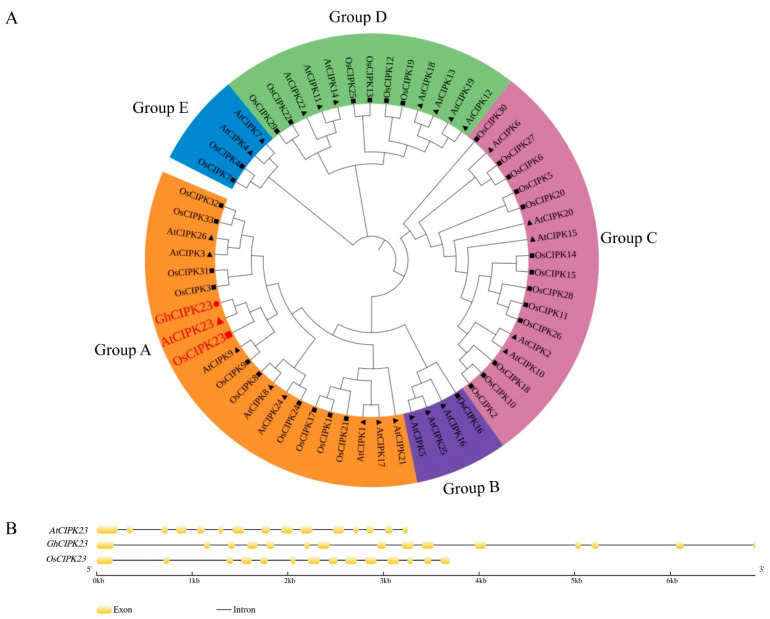
Phylogenetic relationships (**A**) and gene structure (**B**) between GhCIPK23 protein and *CIPK* gene family members in *Arabidopsis* and rice. The symbol ● represents GhCIPK23 in upland cotton. The symbols ▲ and ■ represent members of *Arabidopsis* and rice *CIPK* gene families, respectively, and the symbols in red represent the CIPK23 protein.

**Figure 5 ijms-23-12040-f005:**
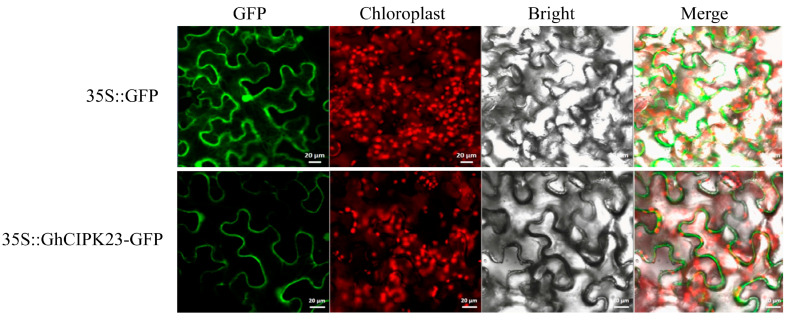
Subcellular localization of GhCIPK23 protein in upland cotton.

**Figure 6 ijms-23-12040-f006:**
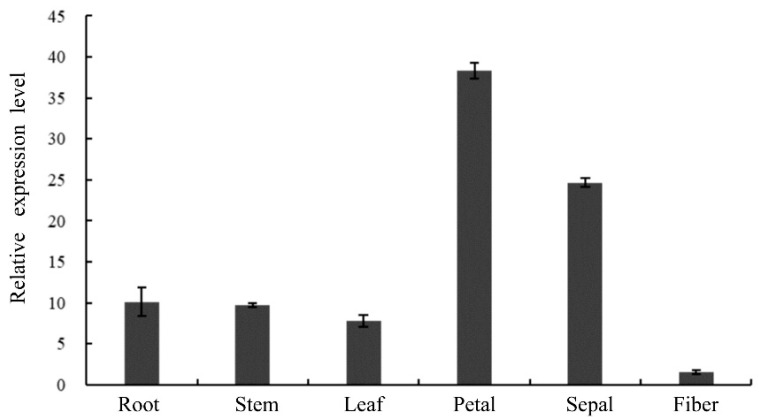
Expression pattern analysis of *GhCIPK23* gene in different tissues of upland cotton. The error bars represent standard errors of three independent repetitions.

**Figure 7 ijms-23-12040-f007:**
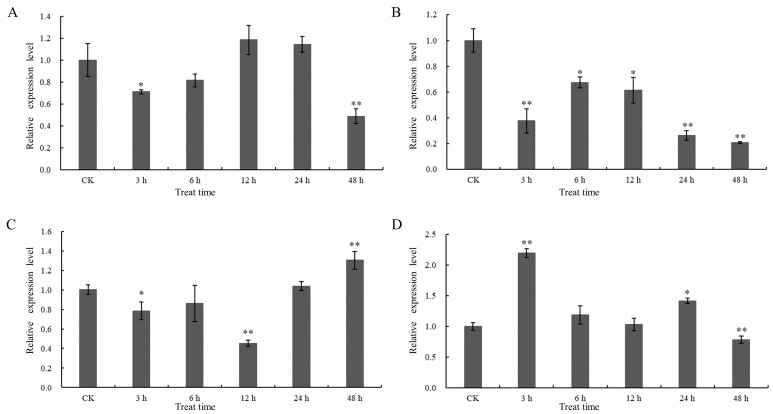
Expression pattern analysis of *GhCIPK23* gene under different stress conditions in upland cotton. (**A**) Drought treatment (20% PEG6000); (**B**) salt (200 mM NaCl); (**C**) low-temperature treatment (4 °C); (**D**) exogenous abscisic acid treatment (100 μM ABA). The error bars represent standard errors of three independent repetitions. * *p* < 0.05; ** *p* < 0.01.

**Figure 8 ijms-23-12040-f008:**
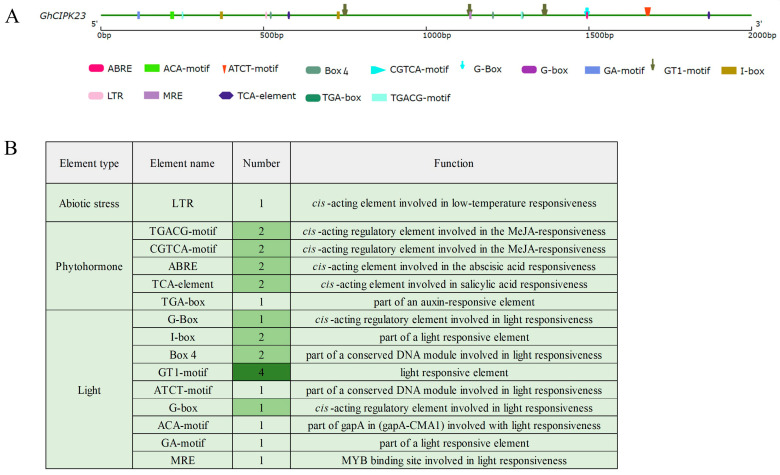
*Cis*-acting elements analysis of *GhCIPK23* promoter in upland cotton. (**A**) The position distribution diagram of *cis*-acting elements, and different colors represent different kinds of *cis*-acting elements. (**B**) Category and number of *cis*-acting elements, and the color from light to dark represents the number of *cis*-acting elements from less to more.

## Data Availability

The data presented in this study are available on request from the corresponding author.

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
