# Peer review of "Sequence Characteristics and Expression Analysis of *GhCIPK23* Gene in Upland Cotton (*Gossypium hirsutum* L.)"

_ijms, 2022, doi:10.3390/ijms231912040_

Round 1
Reviewer 1 Report
Correct the grammar: Total RNA was extracted by the manual of the plant- line 97
Which stage, timings and tissue type was used to generate cDNA from which the whole CDS was amplified and cloned?
Restructure lines 159 to 166
Add high resolution figures of Figure 3A and 3B
As the promoter regions contains higher number of light responsive (14) and phytohormone responsive elements (9), authors are requested to perform a gene expression analysis for these parameters also such as by providing different concentrations of hormones in the media validate the response of these elements.
Author Response
#Reviewer 1
Comment 1: Correct the grammar: Total RNA was extracted by the manual of the plant- line 97
Answer: We have corrected the grammar mistakes of this sentence, and “Total RNA was extracted by the manual of the plant” was changed to “Total RNA was extracted according to the manual of the plant” in our revised manuscript (Line 97).
Comment 2: Which stage, timings and tissue type was used to generate cDNA from which the whole CDS was amplified and cloned?
Restructure lines 159 to 166
Answer: We used the cDNA of the roots of four-leaf stage cotton seedlings of Baimian No.1 as a template, and the CDS of the GhCIPK23 gene was amplified and cloned by PCR. In our revised manuscript, we have added these contents, and re-phrase this paragraph in the result section (lines 160-169).
Comment 3: Add high resolution figures of Figure 3A and 3B
Answer: Figure 3A and 3B were replaced with high resolution figures in our revised manuscript (Figure 3).
Comment 4: As the promoter regions contains higher number of light responsive (14) and phytohormone responsive elements (9), authors are requested to perform a gene expression analysis for these parameters also such as by providing different concentrations of hormones in the media validate the response of these elements.
Answer: Many thanks for reviewer’s helpful suggestions. In hormonal responses, abscisic acid (ABA) plays a key role in plant response to various abiotic stress [1]. Previous studies have demonstrated that the expression of many CIPK genes is responsive to ABA treatment [2-6]. Besides this, many studies also found that the ABA-responsive elements (ABREs) widely exist in the promoter region of CIPK family genes. For example, in honeysuckle, ABREs are found in 13 out of 17 LjCIPK genes [7], which is also found in sugar beet (nearly one half of BvCIPKs is identified to contain ABREs) [8], suggesting that the most CIPK genes might be involved in the ABA signal pathway. Based on this, we have analyzed the expression pattern of GhCIPK23 under ABA treatment, and our results showed that the expression of GhCIPK23 was significantly up-regulated by ABA treatment in upland cotton, which was consistent with the expression pattern of OsCIPK23 [9] and TaCIPK23 [4] under ABA treatment. In accordance with this, two abscisic acid-responsive elements (ABREs) were found in the promoter of GhCIPK23 gene in our study, which may explain why GhCIPK23 expression was induced by ABA.
In our revised manuscript, the result of GhCIPK23 expression under ABA treatment was added in the results section (lines 252-254, Fig.7-D), and a paragraph about CIPK family genes in response to ABA was added in the discussion section (lines 370-391).
References
[1] Vishwakarma K, Upadhyay N, Kumar N, Yadav G, Singh J, Mishra RK, Kumar V, Verma R, Upadhyay R G, Pandey M, Sharma S. Abscisic acid signaling and abiotic stress tolerance in plants: a review on current knowledge and future prospects. Front Plant Sci. 2017, 8:161.
[2] Ma R, Liu W, Li S, Zhu X, Yang J, Zhang N, Si H. Genome-wide identification, characterization and expression analysis of the CIPK gene family in Potato (Solanum tuberosum L.) and the role of StCIPK10 in response to drought and osmotic stress. Int J Mol Sci. 2021, 22(24):13535.
[3] Sun T, Wang Y, Wang M, Li T, Zhou Y, Wang X, Wei S, He G, Yang G. Identification and comprehensive analyses of the CBL and CIPK gene families in wheat (Triticum aestivum L.). BMC Plant Biol. 2015,15:269.
[4] Cui X Y, Du Y T, Fu J D, Yu T F, Wang C T, Chen M, Chen J, Ma Y Z, Xu Z S. Wheat CBL-interacting protein kinase 23 positively regulates drought stress and ABA responses. BMC Plant Biol. 2018, 18(1):93.
[5] Kim K N, Cheong Y H, Grant J J, Pandey G K, Luan S. CIPK3, a calcium sensor-associated protein kinase that regulates abscisic acid and cold signal transduction in Arabidopsis. Plant Cell. 2003, 15(2):411-23.
[6] Zhao J, Yu A, Du Y, Wang G, Li Y, Zhao G, Wang X, Zhang W, Cheng K, Liu X, Wang Z, Wang Y. Foxtail millet (Setaria italica (L.) P. Beauv) CIPKs are responsive to ABA and abiotic stresses. Plos One. 2019, 14(11):e0225091.
[7] Huang L, Li Z, Fu Q, Liang C, Liu Z, Liu Q, Pu G, Li J. Genome-wide identification of CBL-CIPK gene family in honeysuckle (Lonicera japonica Thunb.) and their regulated expression under salt stress. Front Genet. 2021, 12:751040.
[8] Wu G Q, Xie L L, Wang J L, Wang B C, Li Z Q. Genome-Wide identification of CIPK genes in sugar beet (Beta vulgaris) and their expression under NaCl stress. J Plant Growth Regul. 2022:1-15.
[9] Yang W, Kong Z, Omo-Ikerodah E, Xu W, Li Q, Xue Y. Calcineurin B-like interacting protein kinase OsCIPK23 functions in pollination and drought stress responses in rice (Oryza sativa L.). J Genet Genomics. 2008, 35(9):531-43.
Reviewer 2 Report
I congratulate the authors for their work. Characterization of GhCIPK23 was done in upland cotton which helps to explore this gene for upland cotton crop improvement for abiotic stress.
The authors can consider revising the manuscript for minor grammatical errors and questions which are as follows
Line 22
“pression analysis showed that GhCIPK23 was the highest expression in petals, followed by sepals”
pression analysis showed that GhCIPK23 had highest expression in petals, followed by sepals
line 37
“second messengers” can be changed to secondary messengers
“will show” usage is not appropriate.
Line 80
“stress resistance” can be changed to stress tolerance
Line 107 and 108
The PCR program was 94.0 °C for 2 min; 98.0 °C for 20 s and 56.0 °C for1 min 30 s, for 33 cycles; and 72 °C for 10 min.
Do PCR was carried out with an annealing temperature of 56.0 °C for 1 min 30 s without an extension step?
Line 143
“Specific primers were designed as follows:” instead Specific primers were as follows:
If the downregulation of GhCIPK23 under salt stress is attributed to the intrinsic salt tolerance ability of upland cotton the author should have provided the threshold salt concentration above which homeostasis is disturbed in upland cotton (through experimentation or from literature). The authors can explore do potassium regulation can be combined with salt stress as maintaining the sodium-potassium ratio is critical in salt tolerance. More literature can be provided to substantiate the downregulation.
Thank you
Author Response
#Reviewer 2
I congratulate the authors for their work. Characterization of GhCIPK23 was done in upland cotton which helps to explore this gene for upland cotton crop improvement for abiotic stress.
The authors can consider revising the manuscript for minor grammatical errors and questions which are as follows
Comment 1: Line 22
“pression analysis showed that GhCIPK23 was the highest expression in petals, followed by sepals”
pression analysis showed that GhCIPK23 had highest expression in petals, followed by sepals
Answer: Many thanks for reviewer’s helpful suggestions. We have modified this sentence in our revised manuscript (Line 22).
Comment 2: line 3
“second messengers” can be changed to secondary messengers
“will show” usage is not appropriate.
Answer: We have changed “second messengers” to “secondary messengers” in our revised manuscript (line 37).
We have modified “will show” to “will appear” in our revised manuscript (line 37).
Comment 3: Line 80
“stress resistance” can be changed to stress tolerance
Answer: We have changed “stress resistance” to “stress tolerance” in our revised manuscript (line 80).
Comment 4: Line 107 and 108
The PCR program was 94.0 °C for 2 min; 98.0 °C for 20 s and 56.0 °C for1 min 30 s, for 33 cycles; and 72 °C for 10 min.
Do PCR was carried out with an annealing temperature of 56.0 °C for 1 min 30 s without an extension step?
Answer: The PCR program used in our study was a two-step method (annealing and extension was combined into one step) with reference to the instructions and slight modification. In the original manuscript, the description of PCR program was not clear enough, and it has been modified in our revised manuscript (lines107-110), as followers: 94.0 °C (pre-denatured) for 2 min; 98.0 °C (denaturation) for 20 s and 56.0 °C (annealing and extension) for 1 min 30 s, for 33 cycles; and 72 °C (final extension) for 10 min.
Comment 5: Line 143
“Specific primers were designed as follows:” instead Specific primers were as follows:
Answer: We have changed “Specific primers were designed as follows:” to “Specific primers were as follows:” in our revised manuscript (line 144).
Comment 6: If the downregulation of GhCIPK23 under salt stress is attributed to the intrinsic salt tolerance ability of upland cotton the author should have provided the threshold salt concentration above which homeostasis is disturbed in upland cotton (through experimentation or from literature). The authors can explore do potassium regulation can be combined with salt stress as maintaining the sodium-potassium ratio is critical in salt tolerance. More literature can be provided to substantiate the downregulation.
Answer: Based on helpful comments from the reviewer. we have revised the discussion accordingly (lines 354-369) and more literatures supporting down-regulation are provided in our revised manuscript, as follows:
However, our study found that GhCIPK23 showed a different expression pattern under salt stress than under other abiotic stresses and that this gene was significantly down-regulated within the salt stress treatment period. The same results were observed in poplar and wheat [1, 2]. In poplar, the gene expression of PtrCIPK1, PtrCIPK5, PtrCIPK8 and PtrCIPK18 showed an overall downward trend under salt stress [1]. Under high salinity conditions, TaCIPK25 expression was markedly down-regulated in roots; Overexpression of TaCIPK25 resulted in hypersensitivity to Na+, superfluous accumulation of Na+ and decreased salt tolerance in transgenic wheat [2]. Previous studies have shown that, K+/Na+ homeostasis is the primary core response for plant to tolerate salinity [3]. The down-regulated expression of GhCIPK23 might participate in reducing Na+ accumulation and balancing K+/Na+ homeostasis under salt stress in upland cotton. Recent research shows that shown that potassium signaling has been shown to act as a key determinant of plant salinity tolerance [4-7]. But it is unclear whether GhCIPK23 may also involve in K+ uptake by plant roots under high sodium conditions, and the function of GhCIPK23 in regulating Na+ and K+ homeostasis needs to be addressed in the future research.
References
[1] Bai X, Ji J, Wang W, Gu C, Yu Q, Jiang J, Yang C, Liu G. Characterization of CBL-interacting protein kinases’ gene family and expression pattern reveal their important roles in response to salt stress in poplar. Forests. 2022, 13(9):1353.
[2] Jin X, Sun T, Wang X, Su P, Ma J, He G, Yang G. Wheat CBL-interacting protein kinase 25 negatively regulates salt tolerance in transgenic wheat. Sci Rep. 2016, 6:28884.
[3] Zhang H, Feng H, Zhang J, Ge R, Zhang L, Wang Y, Li L, Wei J, Li R. Emerging crosstalk between two signaling pathways coordinates K+ and Na+ homeostasis in the halophyte Hordeum brevisubulatum. J Exp Bot. 2020, 71(14):4345-4358.
[4] Xiao L, Shi Y, Wang R, Feng Y, Wang L, Zhang H, Shi X, Jing G, Deng P, Song T, Jing W, Zhang W. The transcription factor OsMYBc and an E3 ligase regulate expression of a K+ transporter during salt stress. Plant Physiol. 2022, 190(1):843-859.
[5] Song T, Shi Y, Shen L, Cao C, Shen Y, Jing W, Tian Q, Lin F, Li W, Zhang W. An endoplasmic reticulum-localized cytochrome b5 regulates high-affinity K+ transport in response to salt stress in rice. Proc Natl Acad Sci. 2021, 118(50):e2114347118.
[6] Kumari S, Chhillar H, Chopra P, Khanna R R, Khan M I R. Potassium: A track to develop salinity tolerant plants. Plant Physiol Biochem. 2021, 167:1011-1023.
[7] Amo J, Lara A, Martínez-Martínez A, Martínez V, Rubio F, Nieves-Cordones M. The protein kinase SlCIPK23 boosts K+ and Na+ uptake in tomato plants. Plant Cell Environ. 2021, 44(12):3589-3605.
Round 2
Reviewer 1 Report
The authors have addressed all the comments and revised the manuscript accordingly. It can now be accepted for publication